# Response of Potential Indicators of Soil Quality to Land-Use and Land-Cover Change under a Mediterranean Climate in the Region of Al-Jabal Al-Akhdar, Libya

**Jamal Suliman Alawamy** [1,2], **Siva K. Balasundram** [1,*], **Ahmad Husni Mohd. Hanif** [1] **and Christopher Teh Boon Sung** [1]

1   Faculty of Agriculture, Universiti Putra Malaysia, Serdang 43400, Selangor, Malaysia; jamal.alawamy@gmail.com (J.S.A.); husni@upm.edu.my (A.H.M.H.); chris@upm.edu.my (C.T.B.S.)
2   Faculty of Natural Resources and Environmental Sciences, University of Derna, Derna 21881, Libya
*   Correspondence: siva@upm.edu.my

**Abstract:** Conversion of native lands into agricultural use, coupled with poor land management practices, generally leads to changes in soil properties. Understanding the undesirable effects of land-use and land-cover (LULC) changes on soil properties is essential when planning for sustainable land management. This study was conducted in Al Jabal Al Akhdar region, Libya, to assess the effects of land-use and land-cover changes on soil quality inferred by analyzing the relative changes in 17 chemical, physical, and biological soil properties in the upper layer (0–20 cm) of disturbed and undisturbed soil systems. Soil samples were collected from 180 sampling sites with 60 from each of the three types of LULC prevalent in the study area: natural Mediterranean forests (NMF), rainfed agriculture (RA), and irrigated crops (IC). The soil properties of the two agricultural land uses were compared with soil properties under an adjacent natural forest, which served as a control to assess changes in soil quality resulting from the cultivation of deforested land. The results indicate significant reductions in most soil quality indicators under rainfed agriculture as compared to native forest land. Under irrigated agriculture, there were significant changes ($p \leq 0.05$) in most of the soil quality indicators, generally, indicating a significant reduction in soil quality, except for improvement of nitrogen and phosphorus levels due to frequent fertilizer application. Our data support the notion that changes in land use and land cover, in the absence of sustainable management measures, induce deterioration of soil properties and ultimately may lead to land degradation and productivity decline.

**Keywords:** soil quality; land degradation; land use change; Al Jabal Al Akhdar; Libya

## 1. Introduction

As a Mediterranean region, Libya ranks 17th in the world in terms of land area (1.75 million km$^2$). Limited suitable land and lack of water supplies in Libya are critical constraints on agricultural productivity [1]. Cultivated land in Libya accounts for roughly 9% of the country's total area, with arable land covering just 1% [2,3]. With the exception of the sub-humid zone of Al-Jabal Al-Akhdar situated in the northeast on the Mediterranean coast, most Libyan soils are in arid and semi-arid areas, characterized by a lack of water resources, low fertility and poor vegetation. With an annual rainfall of between 400 and 650 mm, the region of Al-Jabal Al-Akhdar is the wettest and greenest part of Libya, supporting some natural Mediterranean forests and highly productive dryland agriculture. The Al-Jabal Al-Akhdar zone accounts for less than 1% of Libya's total area [4], but covers about 90% of the forest cover in the country. The Al-Jabal Al-Akhdar region has been plagued by extreme vegetation destruction over the past four decades, thus reducing the acreage of natural vegetation and increasing land degradation.

Land degradation in the Al-Jabal Al-Akhdar area has been intensified as a result of prolonged drought events and the effects on the natural environment due to human activities, including removal of natural vegetation cover for firewood utilization, overgrazing,

and agricultural expansion without adherence to best practices. Signs of land degradation in the region of Al-Jabal Al-Akhdar include marked decline or complete loss of vegetation cover, accelerated soil erosion, reduced biodiversity, reduced habitat diversity, and declined crop yield and animal productivity [5].

Rain-fed and irrigated agriculture are the predominant agricultural systems in the region. These systems are often managed by smallholder farming families using conventional crop cultivation and animal husbandry techniques. Over the years, these systems have inevitably caused an increase in clearance of natural Mediterranean forest to create more agricultural fields. Aburas and Abdel Rahman [6] reported that the limestone nature of most soils in the north of Jabal Al-Akhdar, the natural vegetation prevailing in the region, and the soil's containment of iron oxides and kaolinite clay minerals contributed greatly to the formation of current soil properties. The cohesion of its construction has reduced the relative differences of soil properties despite the difference in soil depth. However, the conversion of natural vegetation into agricultural fields in this area can lead to adverse effects on the quality of the soil by altering the chemical, physical and biological properties of fragile soils that are already shallow and subject to a decline in organic matter [4,7]. Alawamy et al. [8] investigated the LULC change in Al-Bayda-Lussaitah, Al-Jabal Al-Akhdar, between 1985 and 2017 using time-series Landsat data and reported a significant decline of natural forest and a corresponding agriculture expansion. According to this study, forest land lost 39% of its total area over 32 years while the rainfed and irrigated lands expanded by 55% and 85%, respectively. Changes in land use and inadequate management systems are significantly threatening the productivity and sustainability of soil resources in this area. An in depth understanding of soil quality degradation is necessary to introduce important soil conservation measures in this region of Libya. These efforts must be directly linked to sustainable land-use protocols.

Several studies have investigated the effects of land-use conversion on soil quality in native forest and adjacent cultivated lands using a variety of chemical, physical, and biological soil indicators and highlighted significant influences on most soil properties [9,10]. However, such studies have rarely been conducted in the region of Al-Jabal Al-Akhdar, Libya, where natural forest and agricultural lands are undergoing severe degradation. Despite the widespread awareness of the issue of land degradation in the region of Al-Jabal Al-Akhdar, what have been done mostly focus on soil erosion or studying individual characteristics in isolation from other aspects of land use and management. There have been few local research works that have described soil degradation in the Al-Jabal Al-Akhdar region. The most recent and important studies were that by Abdalrahman et al. [11], Aburas et al. [12], and Abdalrahman and Mossa [13]. The previous studies may give general ideas about the region in terms of location, vegetation cover, soil types, water resources, and geology. However, those studies lack depth and have no comprehensive analysis of soil characteristics. They do not specify and describe the degradation problem in adequate detail and do not investigate the reasons for soil degradation due to deforestation and land-use change.

Improving our understanding of changes in soil characteristics is important for identifying production constraints and essential when recommending effective land-use management practices and interventions for soil conservation and remediation so as to ensure long-term sustainable productivity [14]. Due to the limited available literature on the Al-Jabal Al-Akhdar region concerning the change of soil properties due to conversion of natural forest land into agricultural fields, and also the need for further data to enhance decision support for sustainable development, the current study was aimed at: (1) determining how agricultural practices influence the soil chemical, physical, and biological parameters of the study area, and (2) identifying the possible factors leading to the changes in the parameters of disturbed soil. Such information will help to optimize the use of the land surface by elaborating rational management options and suggesting the relevant remedial and conservation measures that endeavor to maintain essential landscape functions and to sustain productivity.

## 2. Materials and Methods

### 2.1. Study Area

To ensure that comparison of soil quality indicators is carried out under similar conditions, all sampling sites were chosen from the Lussaitah agricultural area (Figure 1), taking into account the history of land use and the homogeneity of soil types, topography, slope, elevation, and climate conditions. Lussaitah is a moderately sloping plain located north of the city of Al-Bayda, Libya (32°46′ to 32°55′ N and 21°31′ to 21°44′ E) at 200–400 m above sea level and separated from the coastal plain by a steep escarpment. The sampling area, characterized by thick Mediterranean forest cover, covered an area of approximately 22,700 ha (Figure 1).

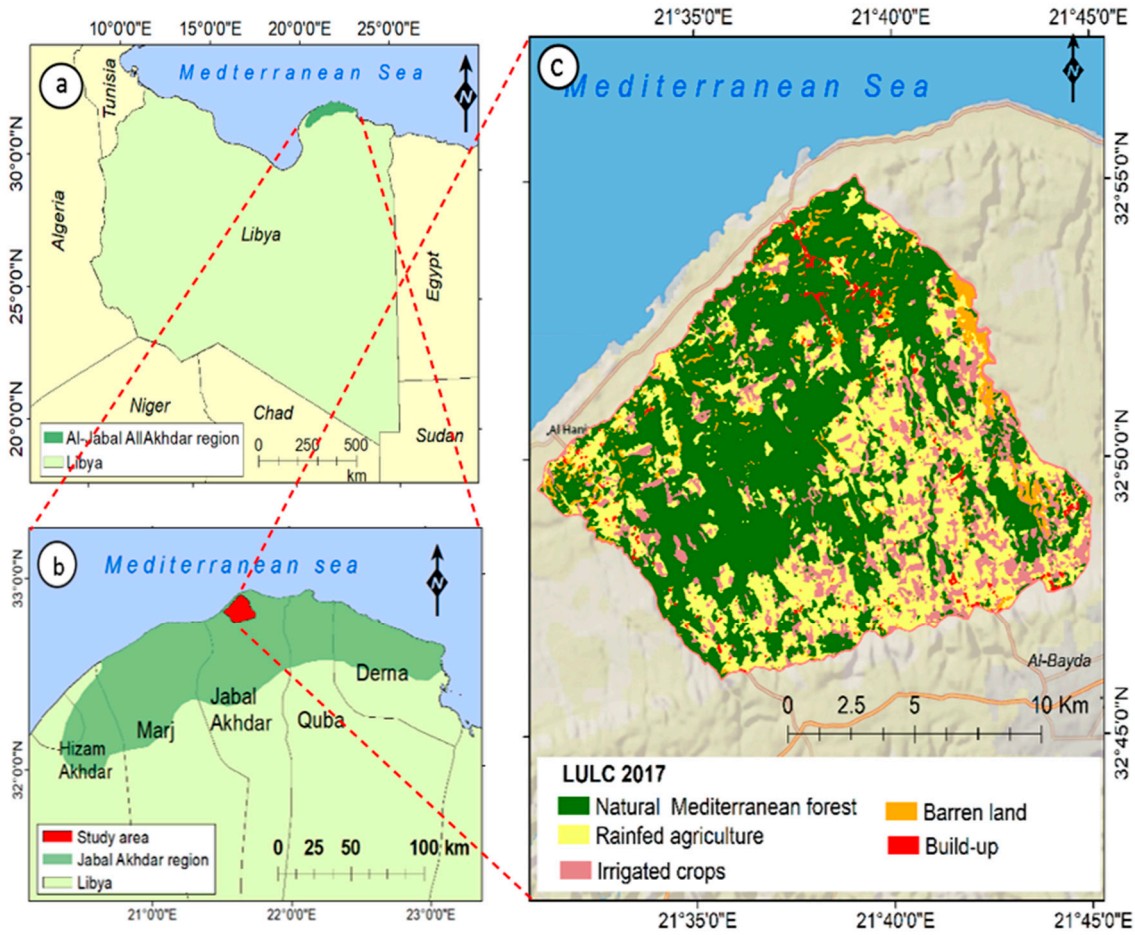

**Figure 1.** (**a**) Location of Al-Jabal Al-Akhdar region in Libya; (**b**) location of study area in Al-Jabal Al-Akhdar region; (**c**) land use/land cover (LULC) map of the study area of Lussaitah [8].

The area of Lussaitah on the first bench of the Al-Jabal Al-Akhdar region is characterized by a Mediterranean climate where maximum temperature (29 °C) occurs between May and June and minimum temperature (14 °C) is observed between January and February [15]. Rainfall is generally irregular and unequally distributed with an annual rainfall of approximately 400 mm during the winter season (October to February), of which about 75% falls between December and January [4].

Soils in the study area were categorized by Selkhoz Prom Export [16] into two overlapping groups according to the Russian classification scheme, i.e., Red Ferrisiallitic and Red Rendzinas which correspond approximately to Lithic Rhodoxeralfs and Lithic Rendolls, respectively, in the soil taxonomy scheme of the United States Department of Agriculture (USDA). Red Ferrisiallitic soils (Rhodoxeralfs) are considered to be the most significant and predominant soils in the study area and also known as Terra Rossa [17].

The study area is located in the evergreen Mediterranean forest zone, where the hemi-xerophilous and shrub vegetation of the Mediterranean predominates and occurs in the form of Maquis under conditions of semi-arid and sub-humid climates [4,16]. Lussaitah is an important economic area within the Al-Jabal Al-Akhdar region, where agricultural activities are centered extensively and is considered the region's most important food basket. Historically, the key activities of the local population in Lussaitah have been extensive dry farming of wheat and barley, in addition to grazing of goats and sheep. At present, the main types of land use found in the study region are rainfed agriculture and irrigated agriculture [4]. Irrigated land is often cultivated as small fields for producing various vegetable crops, including tomato, cucumber, zucchini, cabbage, onion, cauliflower and eggplant Wheat and barley are the primary rainfed crops grown in the study region. In non-cultivated areas dominated by natural vegetation, grazing operations are typically carried out after crop harvest in the cultivated fields [4,18,19].

Unfortunately, agricultural expansion in the study area, and in Al-Jabal Al-Akhdar in general, is carried out randomly in the absence of awareness on the sustainable practices and the dependence on excessive use of chemical fertilizers, particularly, in irrigated agriculture [20]. It was reported that about 83% of farms in the Al-Jabal Al-Akhdar region are completely dependent on chemical fertilizers for their agricultural practices [21]. EL-Barasi et al. [22] stated that most farms in the region are improperly managed, and farmers mostly are unaware of the appropriate methods of fertilizer application, with less than 1% are using fertilizers appropriately. In general, the average amount of chemical fertilizers is 122 kg per hectare while urea, superphosphate, ammonium phosphate, potassium chloride, and potassium nitrate are the common fertilizer types [23].

### 2.2. Soil Sampling and Preparation

In order to achieve the objective of assessing the effects of the LULC changes on soil properties as indicators of soil quality degradation, 12 locations were identified for sampling within the study region. Five representative sampling points (sites) for each of the three dominant LULC types, which are rainfed agriculture (RA), irrigated crops (IC), and natural Mediterranean forests (NMF), were identified within each of the 12 locations resulting in a total of 180 sampling sites (60 sampling sites for each LULC type). On the basis of historical land-use data collected via landholder interviews and historical imagery from Google Earth, sampling sites were selected only in fields cultivated for at least 15 years after conversion from natural forest.

Soil samples were collected from the three LULC types in the summer of 2016 at a depth of 0–20 cm using a soil auger. Soil samples were composited from 5 sub-samples taken from the center, and four corners of 400 m$^2$ plots (20 m × 20 m) established at each sampling site. Stones, gravels, visible roots, and organic residues were manually removed from the soil samples. Additionally, undisturbed soil samples were also collected from each sampling site (180 samples in total) using sharp-edged steel cylinders of 98,125 cm$^3$ volume (5 cm in diameter and 5 cm in height) for bulk density and hydraulic conductivity determinations. For the soil biological analyses, fresh soil samples of about 150 g were collected at each sampling site (180 samples in total). All soil samples were labeled and kept inside plastic bags. The fresh samples were placed in a cooler box to avoid moisture loss, and the samples were then transported to the laboratory for further processing and analyses. In the laboratory, fresh soil samples were stored in a chilled room at 4 °C a few days before conducting biological analysis. The disturbed samples were air-dried at room temperature for three days, ground, and sieved to pass through a 2 mm sieve. Soil water content was determined gravimetrically by mass loss of the disturbed samples during oven-drying at 105 °C for 24 h.

### 2.3. Soil Analysis

To evaluate the effect of LULC change on the soils under study, a set of 17 soil parameters common in earlier investigations as potential indicators of soil quality were quantified.

The parameters quantified were soil organic matter (SOM), soil reaction (pH), electrical conductivity (EC), calcium carbonate ($CaCO_3$), cation exchange capacity (CEC), available potassium ($K_{av}$), available phosphorus ($P_{av}$), and nitrate-nitrogen ($NO_3$-N), particle size distribution (% clay, % silt, and % sand), bulk density (BD), total porosity (TP), macroporosity (MP), available water holding capacity (AWHC), saturated hydraulic conductivity ($K_{sat}$) and basal soil respiration (BSR). SOM was analyzed using the modified Walkley–Black method outlined by Jackson [24]. Soil pH was determined by the potentiometric method in 1:2 soil:water suspension and EC was measured in the water extract using a conductivity meter [24]. $CaCO_3$ content was measured by the gasometric method using a calcimeter [25]. CEC was determined by summation of the exchangeable base cation ($K^+$, $Na^+$, $Ca^{2+}$, $Mg^{2+}$) after extraction with ammonium acetate as described by Jackson [24] where exchangeable $K^+$ and $Na^+$ were quantified in the leachate using a flame photometer, while exchangeable $Ca^{2+}$ and $Mg^{2+}$ were measured by the Versenate titration method. The $P_{av}$ was determined following Olsen [26] using the sodium bicarbonate ($NaHCO_3$) method. The concentration of $NO_3$-N was determined by nitration of salicylate [27]. Particle size distribution was determined by quantifying the relative proportion of clay, silt, and sand in soil samples using the hydrometer method [28]. The BD was determined using the core sampling method described by Klute [29]. TP and MP were calculated following the procedure of Flint and Flint [30]. AWHC was calculated as the difference between moisture content at field capacity and permanent wilting point [31]. The $K_{sat}$ measurements were conducted in the laboratory using the constant-head method described by Klute [29]. BSR was determined by measuring carbon dioxide ($CO_2$) concentration released by microbial biomass during the period of soil incubation in a closed system [32].

### 2.4. Statistical Analysis

The normality assumption was evaluated for each soil property using the Kolmogorov–Smirnov (K–S) test [33]. For the normally distributed data, a parametric assessment (one-way ANOVA) was performed at the probability level ($p$) $\leq$ 0.05 to detect significant differences in the measured variables among the three different LULC types. For the non-normally distributed data, non-parametric Kruskal–Wallis test (followed by the Mann–Whitney U test) was adopted to assess significant differences between the three LULC types [34]. The data were also analyzed using Pearson's correlation to determine the strength of linear relationship between variables.

### 3. Results and Discussion

Soil chemical, physical, and biological properties were analyzed in the laboratory to evaluate the effects of LULC change on land degradation. Table 1 shows the Pearson (r) correlation matrix for the measured soil properties. There was a considerable degree of correlation among the various chemical, physical, and biological properties measured. Correlation analysis of the soil chemical, physical, and biological properties of the investigated area showed a significant relationship ($p < 0.05$) for 98 out of 136 soil attribute pairs (Table 1). Comparisons were conducted for each soil characteristic across three LULC types (NMF, RA, and IC). Statistically significant changes were also identified for evaluating the effects of the different LULC types on the selected soil quality indicators. These findings are discussed below.

**Table 1.** Pearson's correlation coefficients among the measured soil properties.

| | SOM | pH | EC | CaCO$_3$ | CEC | K$_{av}$ | P$_{av}$ | NO$_3$−N | Clay | Silt | Sand | BD | TP | MP | AWHC | K$_{sat}$ |
|---|---|---|---|---|---|---|---|---|---|---|---|---|---|---|---|---|
| pH | −0.09 ns | | | | | | | | | | | | | | | |
| EC | −0.40 ** | −0.24 ** | | | | | | | | | | | | | | |
| CaCO$_3$ | −0.01 ns | 0.37 ** | −0.02 ns | | | | | | | | | | | | | |
| CEC | 0.44 ** | 0.37 ** | −0.27 ** | 0.44 ** | | | | | | | | | | | | |
| K$_{av}$ | 0.53 ** | −0.12 ns | −0.35 ** | −0.11 ns | 0.21 ** | | | | | | | | | | | |
| P$_{av}$ | 0.14 ns | −0.22 ** | 0.52 ** | 0.08 ns | −0.04 ns | −0.08 ns | | | | | | | | | | |
| NO$_3$−N | −0.20 ** | −0.12 ns | 0.54 ** | 0.14 ns | −0.01 ns | −0.19 * | 0.67 ** | | | | | | | | | |
| Clay | −0.62 ** | 0.14 ns | 0.31 ** | 0.21 ** | −0.21 ** | −0.49 ** | 0.15 * | 0.20 ** | | | | | | | | |
| Silt | 0.53 ** | −0.05 ns | −0.29 ** | −0.20 ** | 0.20 ** | 0.50 ** | −0.17 * | −0.24 ** | −0.82 ** | | | | | | | |
| Sand | 0.21 ** | −0.15 * | −0.06 ** | −0.03 ns | 0.03 ns | 0.04 ns | 0.01 ns | 0.03 ns | −0.41 ** | −0.20 ** | | | | | | |
| BD | −0.65 ** | 0.14 ns | 0.35 ** | 0.08 ns | −0.32 ** | −0.48 ** | 0.18 * | 0.17 * | 0.48 ** | −0.46 ** | −0.09 ns | | | | | |
| TP | 0.64 ** | 0.14 ns | −0.33 ** | −0.08 ns | 0.32 ** | 0.47 ** | −0.16 * | −0.16 * | −0.48 ** | 0.45 ** | −0.09 ns | −0.98 ** | | | | |
| MP | 0.58 ** | −0.21 ** | −0.11 ns | −0.11 ns | 0.05 ns | 0.24 ** | −0.08 ns | −0.10 ns | −0.26 ** | 0.23 ** | 0.08 ns | −0.81 ** | 0.82 ** | | | |
| AWHC | 0.62 ** | −0.10 ns | −0.46 ** | −0.06 ns | 0.23 ** | 0.51 ** | −0.29 ** | −0.27 ** | −0.54 ** | 0.48 ** | 0.15 * | −0.61 ** | 0.60 ** | 0.38 ** | | |
| K$_{sat}$ | 0.81 ** | −0.03 ns | −0.46 ** | −0.02 ns | 0.42 ** | 0.58 ** | −0.15 * | −0.29 ** | −0.63 ** | 0.58 ** | 0.15 * | −0.64 ** | 0.63 ** | 0.35 ** | 0.63 ** | |
| BSR | 0.74 ** | −0.07 ns | −0.24 ** | 0.01 ns | 0.25 ** | 0.23 ** | −0.10 ns | −0.10 ns | −0.30 ** | 0.27 * | 0.22 ** | −0.39 ** | 0.38 ** | 0.26 ** | 0.30 ** | 0.40 ** |

* significant at the $p < 0.05$, ** significant at $p < 0.01$, ns = non significant.

### 3.1. Response of Chemical Soil Quality Indicators to Land-Use/Land-Cover (LULC) Change

3.1.1. Soil Organic Matter (SOM)

The LULC changes in the study area have resulted in a substantial decrease in SOM. Significant differences ($p < 0.001$) in SOM were found among the three different LULC types (Table 2). Mean values of SOM were in the order: NMF (4.98%) > RA (3.20%) > IC (2.96%). From Table 2, in comparison to NMF, depletions in SOM under RA and IC were estimated to be 36% and 41%, respectively. Several studies have also reported a significant loss of SOM in cultivated lands after deforestation. For example, Raiesi [35], Lizaga et al. [36], and Willy et al. [37] reported decreases in SOM in cultivated lands ranging from 33% to 72%. These findings highlight the urgent need to improve agricultural management systems in order to maintain and improve the level of SOM in cultivated soil required for land sustainability [38].

The relatively higher SOM under NMF could be attributed to the high organic matter input to the soil as a result of tree leaves, stems, barks, flowers, logs, and fruits [39]. In addition, the amount of microorganisms, animals, and roots in forest land contribute to the increase and conservation of SOM [40,41]. In contrast, cultivation usually depletes SOM content because of the consequent reduction in the above and below-ground organic matter inputs due to crop harvesting with significant removal of residues from plants [35,37] and post-grazing which often occurs in the irrigated and rainfed land after harvesting [42]. In addition, intensive tillage and harrowing cause significant reduction in SOM by reducing the amount of root biomass in surface soils, mixing of organic matter-rich surface soil with organic matter-poor subsurface soil [43], and destruction of soil aggregate which lead to exposing the existing organic matter to microbial decay [43–45]. Accelerated erosion by wind and water caused by lesser vegetation cover is an important factor contributing to the removal of SOM in agricultural land [46]. Lower SOM content in IC relative to RA may be attributable to the slow decomposition rate of grain (wheat and barley) residues in RA, as compared to vegetable residues in IC, and the frequent tillage operations in IC.

**Table 2.** Soil chemical analyses across different land use/land cover.

| Soil Property | Land Use/Cover | Mean | Standard Deviation | Standard Errors | Min. | Max. | *p*-Value |
|---|---|---|---|---|---|---|---|
| SOM (%) | NMF | 4.98 [a] | 0.734 | 0.095 | 3.80 | 6.37 | |
| | RA | 3.20 [b] | 0.627 | 0.080 | 1.79 | 4.23 | <0.01 ** |
| | IC | 2.96 [c] | 0.412 | 0.053 | 2.18 | 3.89 | |
| pH | NMF | 7.93 [a] | 0.193 | 0.025 | 7.50 | 8.25 | |
| | RA | 7.94 [a] | 0.240 | 0.031 | 7.51 | 8.32 | 0.105 [ns] |
| | IC | 8.01 [a] | 0.210 | 0.027 | 7.52 | 8.41 | |
| EC (dS m$^{-1}$) | NMF | 0.24 [a] | 0.044 | 0.006 | 0.15 | 0.35 | |
| | RA | 0.26 [a] | 0.062 | 0.008 | 0.16 | 0.47 | <0.01 ** |
| | IC | 0.54 [b] | 0.138 | 0.018 | 0.31 | 0.98 | |
| CaCO$_3$ (%) | NMF | 0.84 [a] | 0.877 | 0.113 | 0.09 | 4.04 | |
| | RA | 0.98 [a] | 1.134 | 0.146 | 0.18 | 4.46 | 0.349 [ns] |
| | IC | 1.10 [a] | 1.128 | 0.146 | 0.12 | 3.88 | |
| CEC (meq 100 g$^{-1}$) | NMF | 19.93 [a] | 2.598 | 0.335 | 13.70 | 26.09 | |
| | RA | 16.05 [b] | 4.112 | 0.531 | 9.80 | 23.75 | <0.01 ** |
| | IC | 17.03 [b] | 3.422 | 0.442 | 10.45 | 24.15 | |
| K$_{av}$ (meq 100 g$^{-1}$) | NMF | 1.78 [a] | 0.326 | 0.042 | 0.89 | 2.78 | |
| | RA | 1.27 [b] | 0.345 | 0.044 | 0.59 | 2.43 | <0.01 ** |
| | IC | 1.14 [b] | 0.279 | 0.036 | 0.73 | 2.00 | |
| P$_{av}$ (mg L$^{-1}$) | NMF | 8.74 [a] | 3.720 | 0.480 | 2.85 | 18.87 | |
| | RA | 5.27 [b] | 3.533 | 0.456 | 1.23 | 15.23 | <0.01 ** |
| | IC | 16.53 [c] | 6.156 | 0.795 | 9.01 | 33.45 | |
| NO$_3$-N (mg L$^{-1}$) | NMF | 12.67 [a] | 3.369 | 0.435 | 7.02 | 23.60 | |
| | RA | 12.21 [a] | 3.599 | 0.465 | 7.26 | 21.60 | <0.01 ** |
| | IC | 18.33 [b] | 5.021 | 0.648 | 9.96 | 27.43 | |

ns = not significantly different at $p < 0.05$; ** significantly different at $p < 0.01$. Mean values followed by a different superscript are significantly different at $p < 0.01$.

### 3.1.2. Soil Reaction (pH)

Soil pH in the study area ranged from slightly to moderately alkaline [47] with values ranging from 7.50 to 8.41 (Table 2). The alkalinity of the soils is due to the dominance of alkaline parent material, along with the presence of base-forming cations associated with carbonates and bicarbonates found naturally in these soils. Contrary to expectations, there was no significant difference in soil pH across LULC. The results from this study are in agreement with Aburas [7] and Willy et al. [37], who reported a non-significant effect in soil pH after conversion of forest land into agriculture.

### 3.1.3. Electrical Conductivity (EC)

Results showed low EC values with an overall mean value of 0.35 dS m$^{-1}$ for the whole study area (Table 2). The EC mean values for the three LULC types were in the order: IC (0.54 dS m$^{-1}$) > RA (0.26 dS m$^{-1}$) $\geq$ NMF (0.24 dS m$^{-1}$) which differed significantly at $p < 0.01$ (Table 2), indicating that transforming forest land to agricultural land impacts soil salinity. In comparison to NMF, soil EC in IC and RA increased by 124% and 8%, respectively.

Variation in EC across LULC is possibly due to the frequent addition of chemical fertilizers in soils under IC, as compared to RA (fewer fertilizer applications) and NMF (complete absence of fertilizer application) [48]. Another factor that contributed to accumulation of salts in soils under IC is the large dependence on groundwater, followed by the adoption of drip irrigation. Additionally, lower vegetation cover in cultivated land, which can lead to higher temperatures and evaporation rates, could have possibly caused an increase in soil salinity [48]. Salinity is likely the most significant abiotic factor that restricts plant development with significant decreases in above and below ground biomass and has a strong unfavorable influence on the soil microbiological activity in arid and semi-arid

regions [49]. Despite the relatively higher EC values in soils under IC, it is still way below the detrimental threshold of 4 dS m$^{-1}$.

### 3.1.4. Calcium Carbonate Content (CaCO$_3$)

CaCO$_3$ content in soils of the study area ranged from 0.09% to 4.46%, with a mean value of 0.97% (Table 2). These results are in agreement with those obtained in an earlier study in the region of Al-Jabal Al-Akhdar [50]. Statistically, CaCO$_3$ content within the upper horizon (0–20 cm) of the study area revealed no significant influence of LULC changes on soil CaCO$_3$.

Under long-term irrigated cultivation and semi-arid climatic conditions and irrigation with water high in bicarbonate and sodium ions, low CaCO$_3$ content in soils may lead to a decrease in the portion of ionic calcium (Ca$^{2+}$) and an increase in exchangeable sodium (Na$^+$), which is an undesirable sign of degradation, structural breakdown, and productive capacity decline [51].

### 3.1.5. Cation Exchange Capacity (CEC)

Soil CEC in the study area ranged from 9.80 to 26.09 meq 100 g$^{-1}$ and averaged at 17.67 meq 100 g$^{-1}$ (Table 2). The highest CEC was recorded in soils under NMF with 19.93 meq 100 g$^{-1}$, which differed significantly from the other LULC types. The lowest CEC was recorded in RA (16.08 meq 100 g$^{-1}$), which was not significantly different from that recorded under IC (17.03 meq 100 g$^{-1}$). These findings are in agreement with those obtained by Aburas [7] and Rahmanipour et al. [52], where CEC in agricultural land was lower than that measured in soils under natural vegetation cover.

Generally, the capacity of the soil to maintain exchangeable cations is mainly determined by SOM, pH, quantity and type of clay, and degree of leaching. Clayey textured soil is generally characterized by high CEC values [53] which range between 30–60 meq 100 g$^{-1}$ [47]. Accordingly, the CEC of the clay soil in the study area is relatively low than other clay-textured soils. This can be attributed to the mineralogy of the clay soils, which comprised predominantly layer-silicate minerals [51]. With SOM as the primary source of CEC in soils with a low percentage of permanent-charge clay minerals, SOM decline, particularly under intensive farming practices, can significantly decrease the CEC of these soils.

CEC is usually expected to increase in cultivated soils due to the increase in fertilizer applications. However, in the current study, the CEC of soil under the natural forest cover was significantly greater than the other LULC types. This is expected following the SOM trends in the soils. In addition, CEC levels may be affected by the current continuous cropping. On the other hand, under semi-arid conditions, soil erosion leads to the elimination of the finest and most fertile fractions of the soil [54,55], which results in a decline in CEC and soil productivity. It is important to note that in response to plant nutrient absorption and agricultural practices such as irrigation, fertilization, liming, the addition of organic manures, and others, CEC and ratios of exchangeable cations on the colloidal surface are not constant but rather dynamic [56].

### 3.1.6. Available Potassium (K$_{av}$)

Concentration of soil K$_{av}$ in the study area ranged between 0.59 meq 100 g$^{-1}$ and 2.78 meq 100 g$^{-1}$ and averaged at 1.40 meq 100 g$^{-1}$ (Table 2), indicating a high availability of K. Results show that land-use change and cultivation of deforested land led to a significant decrease ($p < 0.01$) in the concentration of K$_{av}$. NMF recorded a significantly higher concentration of K$_{av}$ (1.78 meq 100 g$^{-1}$) in comparison to the other LULC types.

The higher concentration of K$_{av}$ under NMF could be attributed to the continuous litter fall and nutrient transformation due to vegetation [39]. On the other hand, the decrease in K$_{av}$ concentration under agricultural land use (RA and IC) was expected due to the high removal of K by the continuous cropping and erosion losses. Although K fertilization is common under IC, K$_{av}$ concentration was not significantly different from that under RA. This could be attributed to continuous and intensive vegetable cultivation, particularly

tomato, which consumes large amounts of K during the growing season, in addition to the leaching of K due to irrigation [57,58].

### 3.1.7. Available Phosphorus ($P_{av}$)

Concentration of soil $P_{av}$ in the study area ranged from 1.23 mg $L^{-1}$ under NMF to 33.45 mg $L^{-1}$ under IC and averaged at 10.18 mg $L^{-1}$ (Table 2). The highest $P_{av}$ was recorded in soils under IC while the lowest $P_{av}$ was recorded in soils under RA.

$P_{av}$ concentration in soils under NMF was below optimal and much lower than that in IC. This could be attributed to the naturally deficient state of P due to fixation by $CaCO_3$, which intensifies under an alkaline soil regime. The accumulation of litter in forest soils and the increased availability of materials such as lignin, however, contribute to increasing the availability of P in soil [59]. This, taken together with P removal via crop harvesting [60], explains the relatively higher concentration of $P_{av}$ in soils under NMF as compared to soils under RA.

Agricultural practices that entail timely application of fertilizers have been shown to improve the concentration of $P_{av}$ [37,61]. Our results seem to support this view, where the concentration of $P_{av}$ increased by 89% in the topsoils under IC as compared to NMF. These results are also consistent with an earlier study conducted in the Al-Jabal Al-Akhdar region by Eldiabani [51]. In the current study, the concentration of $P_{av}$ under RA was high ($P_{av} \geq 20$) in 22% of investigated sites and was even above the optimal level for most field crops, i.e., 30 mg $L^{-1}$ [62], in some sites. This suggests that the application of P fertilizers in these sites exceeded plant-soil demands, which may have resulted in additional costs and possible adverse effects on plant growth [63]. These problems could have been avoided if soil tests were performed periodically to enable optimized P fertilization.

### 3.1.8. Nitrate Nitrogen ($NO_3$-N)

Concentration of soil $NO_3$-N varied between 7.02 mg $L^{-1}$ and 27.43 mg $L^{-1}$ and averaged at 14.40 mg $L^{-1}$. The highest $NO_3$-N was recorded in soils under IC (18.33 mg $L^{-1}$) which was significantly different from those obtained under NMF (12.67 mg $L^{-1}$) and RA (12.21 mg $L^{-1}$) (Table 2). The concentration of $NO_3$-N under IC increased by 40% compared to NMF. This considerable increase can be attributed to the large and frequent N fertilizer applications, which are common in an irrigated agriculture system. The current study showed that soil $NO_3$-N in 17% of investigated sites of IC exceeded the optimal level of N for most crops, i.e., 25 mg $L^{-1}$ [62]. Application of N fertilizers is usually needed to maximize crop production; however, when the N application exceeds the crop demand, N accumulates in the soil in the form of nitrate [64]. This accumulation can adversely affect soil quality because it can lead to soil nutrient imbalance and increased $NO_3$-N loss via leaching below the crop root zone [65]. In addition, some crops, particularly leafy vegetables, can accumulate high levels of $NO_3$-N, which can pose serious health hazards when consumed by humans [66].

### 3.2. Response of Physical Soil Quality Indicators to LULC Change

### 3.2.1. Soil Particle Size Distribution

Irrespective of land-use type, soils in the study area exhibited predominance of clay (Table 3). The highest content of clay, silt and sand was observed in soils under IC (62.43%), NMF (40.88%) and RA (30.12%), respectively. Based on the USDA textural classification system, the majority of soils in the study area were classified as clay loam. The dominance of clay textural class throughout the study area indicates the homogeneity of soil-forming processes which is attributable to parent materials rich in clay minerals [4,16,18].

**Table 3.** Soil physical analyses across different land use/land cover.

| Soil Property | Land Use/Cover | Mean | Standard Deviation | Standard Errors | Min. | Max. | *p*-Value |
|---|---|---|---|---|---|---|---|
| Clay (%) | NMF | 46.36 [a] | 3.360 | 0.434 | 38.38 | 51.73 | |
| | RA | 53.94 [b] | 3.633 | 0.469 | 44.98 | 61.28 | <0.01 ** |
| | IC | 54.79 [b] | 3.317 | 0.428 | 46.91 | 62.43 | |
| Silt (%) | NMF | 33.45 [a] | 3.457 | 0.446 | 27.66 | 40.88 | |
| | RA | 26.80 [b] | 3.242 | 0.419 | 19.58 | 34.92 | <0.01 ** |
| | IC | 26.11 [b] | 3.608 | 0.466 | 16.76 | 34.04 | |
| Sand (%) | NMF | 20.19 [a] | 1.722 | 0.222 | 16.96 | 25.61 | |
| | RA | 19.26 [a] | 3.605 | 0.465 | 12.35 | 30.12 | 0.097 [ns] |
| | IC | 19.09 [a] | 3.293 | 0.425 | 11.50 | 25.52 | |
| BD (g cm$^{-3}$) | NMF | 1.19 [a] | 0.070 | 0.009 | 1.09 | 1.35 | |
| | RA | 1.33 [b] | 0.086 | 0.011 | 1.12 | 1.44 | <0.01 ** |
| | IC | 1.36 [b] | 0.081 | 0.010 | 1.18 | 1.50 | |
| TP (%) | NMF | 54.91 [a] | 2.621 | 0.338 | 49.01 | 58.79 | |
| | RA | 49.54 [b] | 3.299 | 0.426 | 45.17 | 57.55 | <0.01 ** |
| | IC | 48.68 [b] | 3.043 | 0.393 | 43.89 | 55.25 | |
| MP (%) | NMF | 19.74 [a] | 3.088 | 0.399 | 12.71 | 25.61 | |
| | RA | 17.50 [b] | 3.424 | 0.442 | 10.76 | 24.72 | <0.01 ** |
| | IC | 16.94 [b] | 3.390 | 0.438 | 10.17 | 23.21 | |
| AWHC (%) | NMF | 17.08 [a] | 2.546 | 0.329 | 10.97 | 23.29 | |
| | RA | 14.24 [b] | 1.891 | 0.244 | 10.97 | 19.01 | <0.01 ** |
| | IC | 12.31 [c] | 1.170 | 0.151 | 9.19 | 17.02 | |
| K$_{sat}$ (cm hr$^{-1}$) | NMF | 6.87 [a] | 0.989 | 0.128 | 4.95 | 8.32 | |
| | RA | 3.66 [b] | 0.905 | 0.117 | 2.49 | 5.88 | <0.01 ** |
| | IC | 3.34 [b] | 0.818 | 0.106 | 1.75 | 4.77 | |

ns = not significantly different at $p < 0.05$; ** significantly different at $p < 0.01$. Mean values followed by a different superscript are significantly different at $p < 0.01$.

Although soil texture and particle size distribution are less dynamic, some changes may occur due to anthropogenic activities, land-use patterns, and management practices [67]. This is in agreement with the results obtained in this study, where significant difference between soil fractions across different LULC types was recorded (Table 3). Clay content was significantly higher under IC (54.79%) and RA (53.94%), compared to NMF (46.36%). Silt content under NMF (33.45%) was significantly higher than those of RA and IC. However, there was no significant difference in sand content across LULC types. In general, our findings conformed to those obtained in the same region of Al-Jabal Al-Akhdar by Gebril [19] and Aburas [7], who reported an increase in clay content and a corresponding decrease in silt content for cultivated soils.

Unlike forest land, frequent plowing practices in cultivated land may lead to a relative change in the sequence of natural soil horizons by disturbing and mixing the upper part of the argillic subsurface horizon with the surface soil layer, which leads to higher clay content in the surface soil. Additionally, the protection provided by the forest vegetation cover would minimize surface erosion, while soil loss due to erosion from agricultural soils will have the effect of moving clay-enriched substance from the argillic horizon closer to the surface [7]. In addition, intensive tillage practices that speed up soil weathering processes and subsequently convert silt fractions to clay could be a possible cause of higher clay content in cultivated land [68]. On the other hand, the lower clay content in soils under NMF might be attributable to the translocation of fine clay particles from the upper to lower horizon under the steady conditions of undisturbed natural land with suitable structure, deep roots, and faster infiltration rates. This is also consistent with previous findings by Shrestha et al. [69], who reported accelerated clay movement from the topsoil to the subsoil under forest land.

### 3.2.2. Bulk Density (BD)

The maximum value of BD was observed in soils under IC (1.50 g cm$^{-3}$) and the minimum was recorded in soils under NMF (1.09 g cm$^{-3}$). Statistically, BD was significantly affected by LULC change (Table 3). The lowest mean BD was found in NMF with 1.19 g cm$^{-3}$, which differed significantly from the other two agricultural land uses (RA and IC). Conversely, the highest mean BD was obtained in soils under IC (1.36 g cm$^{-3}$), which did not differ significantly from that under RA (1.33 g cm$^{-3}$). Similarly, previous studies have reported that BD tends to be higher in soils under agricultural land use than in soils under natural vegetation cover [70,71].

It is known that the BD value of 1.39 g cm$^{-3}$ in clay soils is the initiation point for the restriction of plant root extension and values greater than 1.47 g cm$^{-3}$ are regarded as a limiting factor for root extension [72]. Accordingly, BD in about 32% and 45% of soil sampling sites under RA and IC, respectively, exceeded the root restriction initiation level of 1.39 g cm$^{-3}$ while all soils under NMF were below the critical level with the majority falling within the favorable range of 0.9–1.3 g m$^{-3}$ for plant growth in fine to medium textured soils as classified by Reynolds et al. [73].

Higher BD in agricultural soils is due to machinery traffic during cultivation which led to soil compaction, one of the significant challenges in agricultural fields in several parts of the world [71,74,75]. Higher BD values in the cultivated land and lower values in natural forest land can be related to the level of soil organic matter. A strong negative correlation between BD and SOM was highlighted in previous works [76,77]. The BD of soils in this study was negatively correlated with the soil organic matter content (r = −0.65; $p < 0.01$), indicating the vital role of organic matter in improving soil structure and reducing bulk density (Figure 2). In addition, grazing pressure may be another reason for the occurrence of soil compaction in cultivated lands [78,79].

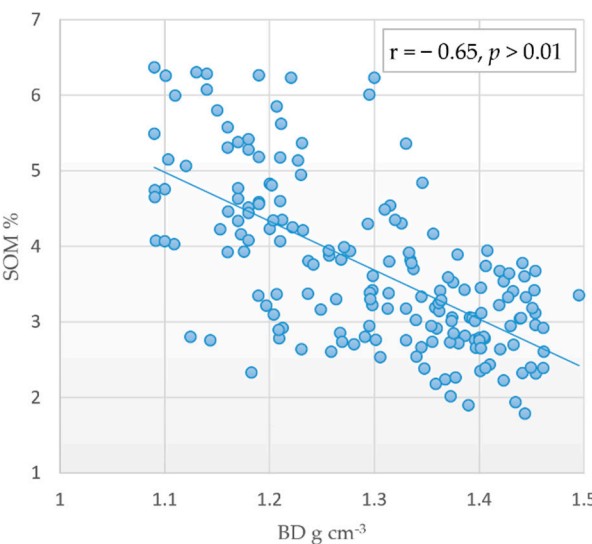

**Figure 2.** Correlation between soil bulk density and soil organic matter content.

### 3.2.3. Soil Porosity

Total porosity (TP) of the soils in the study area ranged from 43.89% to 58.79% and averaged at 51.04% (Table 3). The highest TP was measured in soils under NMF while the lowest was recorded from soils under IC. Macroporosity (MP) of the soils in the study area varied between 10.17% and 25.61% and averaged at 18.06% (Table 3). As with TP, the highest MP was measured in soils under NMF while the lowest was recorded from soils under IC. Results indicate that soils in all sites investigated had a percentage of macro-pores exceeding 10%, which indicates a low level of air-filled porosity [47].

Converting natural land to cultivated land uses in the investigated area led to a significant decrease in soil porosity. Higher soil porosity under NMF is most probably due

to the higher content of SOM, which is critical for preserving sound soil structure via its positive role in forming stable soil aggregates that improves soil porosity and supports movement and availability of air and water to the plant [45]. Moreover, soil compaction due to the use of heavy machinery and summer grazing in cultivated land (RA and IC) is another reason for the decrease in soil porosity [78,79]. On the other hand, clay accumulation in the top plow layer could reduce soil porosity. The solid connectivity between clay particles minimizes the pore size, contributing to less space between the pores [80].

### 3.2.4. Available Water Holding Capacity (AWHC)

AWHC of soils in the study area ranged from 9.19% under IC to 23.29% under NMF and averaged at 18.6%. As shown in Table 3, soils under NMF had the highest mean AWHC (17.08%), while soils under IC recorded the lowest mean (12.31%). AWHC was more sensitive to change under IC than RA where their reductions were estimated at 17% and 28%, respectively, relative to NMF. These findings are in agreement with findings by Irshad et al. [39] and Tesfahunegn [81], where AWHC of the topsoil was shown to be lower in the cultivated soils as compared to forest land.

Although AWHC is one of the most important indicators of soil quality, it is dependent upon other soil properties, such as porosity, texture, bulk density, surface crusts, and organic matter [82]. Variations in AWHC among the different LULC types under comparison in this study could be associated with the difference in organic matter and clay contents of soils. It is well known that AWHC is positively correlated with SOM because organic matter raises soil field capacity above the permanent wilting point [83]. Our findings showed a significant positive relationship (r = 0.62; $p < 0.01$) between AWHC and SOM (Figure 3). With increasing clay content, soils require a higher organic matter content to maintain a given value of aggregate stability [84]. This may explain the decrease in AWHC following the conversion of forest land into farmland due to the increase in clay content accompanied with SOM reduction.

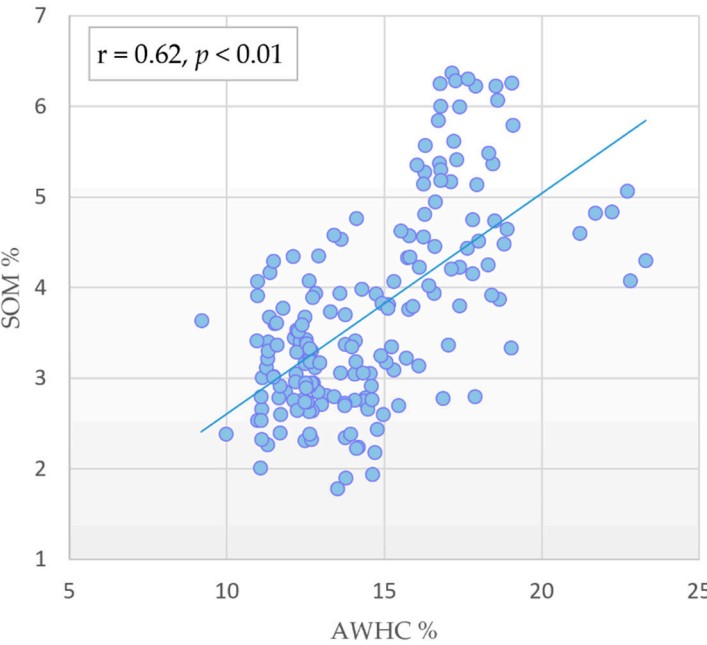

**Figure 3.** Correlation between available water holding capacity and soil organic matter content.

### 3.2.5. Saturated Hydraulic Conductivity (K$_{sat}$)

Topsoil K$_{sat}$ under the three different LULC types varied between 1.75 cm hr$^{-1}$ and 8.32 cm hr$^{-1}$ with the highest value in soils under NMF and the lowest in soils under RA. Topsoil K$_{sat}$ data across the three land-use types were significantly different (Table 3). The reduction in topsoil K$_{sat}$ under RA and IC was estimated at 47% and 51%, respectively, in

comparison to NMF, which indicate the sensitivity of this indicator to LULC change. Our results match those observed in other studies [7,85,86], which confirms that topsoil $K_{sat}$ tends to be greater in forest land than in cultivated soils.

Higher topsoil $K_{sat}$ under the NMF can be attributed to the greater SOM content, which contributes to good topsoil structure and stability than in cultivated soils. This was confirmed by the significant correlation (r = 0.81; $p < 0.01$) between $K_{sat}$ and SOM (Figure 4a). Lower $K_{sat}$ in cultivated soils (RA and IC) could be due to the use of heavy machinery which can affect the continuity of macropores within the topsoil, reduce aggregate stability, and cause soil compaction leading to reduction in the ability of the soil to transmit water from soil surface to below ground [71,74,75,87]. In cultivated soils, $K_{sat}$ can also be reduced due to the compounding effects of rainfall and surface runoff in the absence of vegetation cover [88].

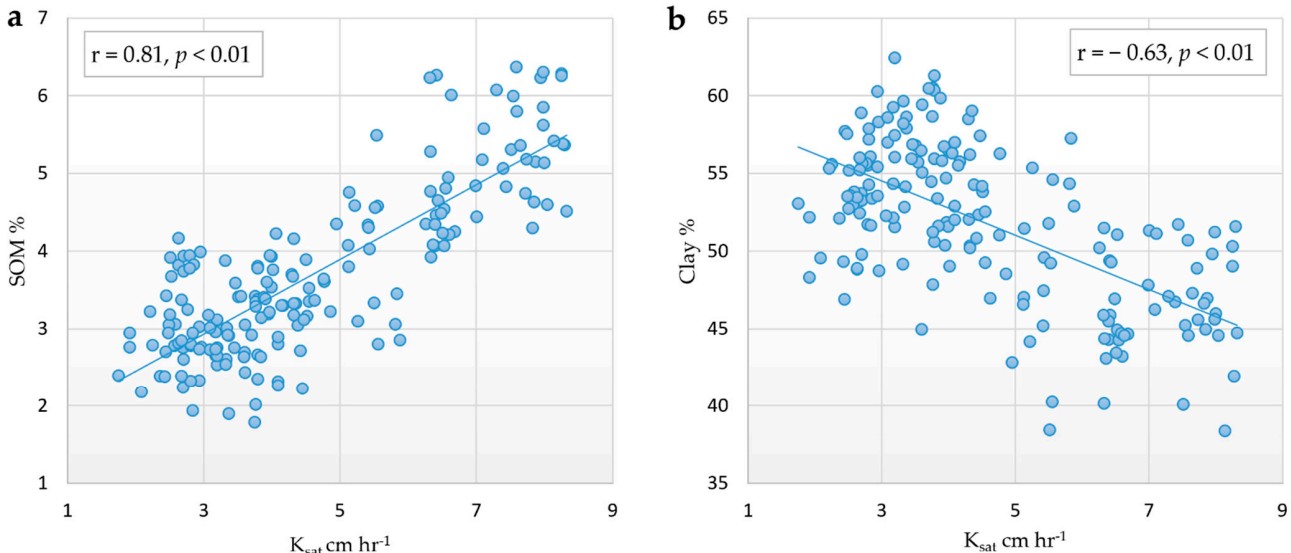

**Figure 4.** Correlation between saturated hydraulic conductivity and soil organic matter content (**a**) and clay content (**b**).

On the other hand, soil texture greatly influences soil permeability and infiltration rate. High sand content promotes faster infiltration, whereas high clay content slows down infiltration due to the stronger influence of clay on the viscosity coefficient and soil water suction compared to coarser soil fractions, even with similar pore sizes [89]. Naturally, the strong connectivity between clay particles reduces the pore size, which contributes to reduced infiltration rate [80]. This was confirmed by the current study, where a significant negative correlation was found between the $K_{sat}$ and clay content (r = −0.63, $p < 0.01$) (Figure 4b).

### 3.3. Response of Biological Soil Quality Indicator to LULC Change

Soil respiration is one of the main divers of soil fertility [90], and has been widely employed as an indicator to track soil quality and degradation due to land-use and land-cover changes [9,10]. BSR refers to the emission of carbon dioxide ($CO_2$) from the surface of the soil into the atmosphere through the decomposition of organic matter by organisms in the soil.

BSR rates varied from 0.05 to 0.34 mg $CO_2$ $g^{-1}$ $day^{-1}$, averaging at 0.17 mg $CO_2$ $g^{-1}$ $day^{-1}$. BSR rates were significantly different across NMF, RA and IC. The highest mean of BSR was recorded under NMF (0.20 mg $CO_2$ $g^{-1}$ $day^{-1}$), and the least was measured under IC (0.15 mg $CO_2$ $g^{-1}$ $day^{-1}$) (Table 4). Reduction in soil respiration under agricultural land use as compared to natural systems has also been reported in previous studies [91,92].

**Table 4.** Basal soil respiration (BSR, mg $CO_2$ g$^{-1}$ day$^{-1}$) across different land-use/land cover types.

| Land Use/Cover | Mean BSR | Standard Deviation | Standard Errors | Min. | Max. | *p*-Value |
|---|---|---|---|---|---|---|
| NMF | 0.20 [a] | 0.051 | 0.007 | 0.10 | 0.34 | |
| RA | 0.17 [b] | 0.055 | 0.007 | 0.08 | 0.28 | <0.01 ** |
| IC | 0.15 [c] | 0.036 | 0.005 | 0.05 | 0.26 | |
| Study area | 0.17 | 0.052 | 0.004 | 0.05 | 0.34 | - |

** significantly different at *p* < 0.01. Mean values followed by a different superscript are significantly different at *p* < 0.01.

Results shown in Tables 2 and 4 indicate that SOM is an important factor in determining the BSR rate in soils. Our findings show a strong positive correlation between SOM content and BSR rate (r = 0.74; *p* < 0.01) (Figure 5). These results are in agreement with that of Mallik and Hu [93], who showed a strong correlation between soil organic matter and microbial respiration during an incubation study. Agricultural management practices can either improve or reduce soil respiration. Leaving crop residues on the soil surface, adopting minimal tillage, using cover crops, or other practices that add organic matter will enhance soil quality and long-term soil respiration. Conversely, tillage practices that end up removing, burying, or burning crop residues result in minimizing organic matter content, which adversely affects the biological activity of soil, leading to reduced soil respiration over the long term [72]. In forest ecosystems, vegetation improves the rates of microbial respiration mainly due to increased organic matter through continual litterfall and root turnover and production of plant detritus, which feeds soil organisms [94].

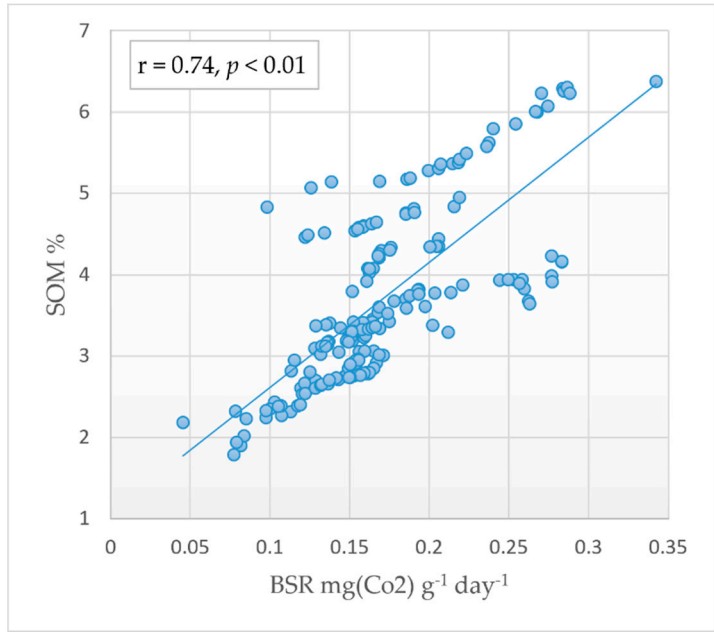

**Figure 5.** Correlation between basal soil respiration and soil organic matter content.

On the other hand, fertilizers may enhance root growth and serve as a food source to microorganisms. Nevertheless, excess dosage of fertilizers may become lethal to the microorganisms responsible for the soil respiration due to increases in pH or salinity [95–98]. Lin et al. [99] reported that overuse of both organic and chemical fertilizers in vegetable cultivation led to degradation of microbiological activities in soil due to soil acidification and accumulation of salts. In the current study, there was a weak correlation between BSR and soil EC (r = −0.24, *p* < 0.01). A weak correlation was also recorded between BSR and clay fraction (r = −0.30, *p* < 0.01), and between BSR and silt fraction (r = 0.27, *p* < 0.01). Medium-textured soils are often favorable to soil respiration because of their

good aeration and higher water availability. Conversely, in clay soils, a sizeable amount of SOM is protected from decomposition by clay particles and other aggregates limiting soil respiration and the associated organic matter mineralization [72,100].

### 4. Conclusions

This study revealed that the conversion of natural land into cultivated farmlands led to a decline in the majority of soil parameters. Compared to natural forest land (NMF), all the soil quality indicators, with the exception of pH, $CaCO_3$ and sand, were significantly reduced under agricultural land use (RA and IC). However, $P_{av}$ and $NO_3$-N under IC was significantly increased due to frequent fertilizer application. The concentration of $P_{av}$ and $NO_3$-N were found to exceed the optimal level for most crops under IC, which may lead to adverse effects on plant growth and increase crop production costs. The decline in soil quality parameters under agricultural land use is probably due to continuous cultivation and intensive practices such as repeated tillage, harrowing operation, inappropriate fertilization, and post-grazing upon crop harvest, in addition to the low turnover of organic matter in soil due to the lower vegetation cover.

Identifying and monitoring LULC change and understanding their impacts is critical to improving the sustainability of soil and land management. To preserve the natural environment, there is an urgent need to develop sound, balanced and sustainable planning for land uses, and then to develop a strategy to re-use agricultural land and marginal natural vegetation land without harming the quality and environment of soil.

**Author Contributions:** Conceptualization, J.S.A. and S.K.B.; methodology, J.S.A. and S.K.B.; software, J.S.A.; data collection, J.S.A.; analysis and validation, J.S.A. and S.K.B.; writing—review and editing, J.S.A. and S.K.B.; supervision, S.K.B., A.H.M.H. and C.T.B.S. All authors have read and agreed to the published version of the manuscript.

**Funding:** This study was funded by the Ministry of Higher Education, Libya, through a Ph.D. grant (293/2013).

**Institutional Review Board Statement:** Not applicable.

**Informed Consent Statement:** Not applicable.

**Data Availability Statement:** Data sharing not applicable.

**Acknowledgments:** We are incredibly indebted to Kamal Abdulsalam, the former head of Soil and Water Department, Omar Al-Mukhtar University, for his invaluable assistance, support, and facilitation of conducting laboratory analyses in Libya. Special appreciation also goes to Mouhamed Abdul Nabi for his valuable help during the field and laboratory investigations. We would also like to thank Murad Abu Ras and Youssef Abdel Rahman for their cooperation during the selection and exploration of the study area.

**Conflicts of Interest:** The authors declare no conflict of interest.

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
