# Peer review of "Response of Potential Indicators of Soil Quality to Land-Use and Land-Cover Change under a Mediterranean Climate in the Region of Al-Jabal Al-Akhdar, Libya"

_sustainability, doi:10.3390/su14010162_

Round 1
Reviewer 1 Report
Dear authors,
It is with great and high sense of responsibility that I write to commend your efforts in putting this intellectual work together. There is no doubt or question about the contribution of the article to knowledge. Notwithstanding, there are areas of improvement as suggested below.
- the arguments in the background are strong but could be improved with recent literature. This also applies to other components of the document.
- the research methods can benefit from clarity and separation of thought in paragraphs. For example, the estimation or analytical procedures are mopped up. This should be separated in line with the objective of the study.
- I suggest you should avoid repetition in the results section.
- Please avoid allowing tables split. If tables must split to other page, it must be a separate table carrying a continuation title. e.g Table 2 Continues
- The results and discussion sections are filled with relatively odd references. It is important to update these sections with more recent references
- Please prepare the manuscript in line with the journal's guideline
- The current conclusion can be upscale with forward looking recommendation.
Author Response
Response as attached
Reviewer 2 Report
Review for “sustainability-1494164”
Response of potential indicators of soil quality to land use and land cover change under a Mediterranean climate in the region of Al-Jabal Al-Akhdar, Libya
This study tried to compare the effects of land use changes on soil properties. The study did not design appropriately and the results need to be reformulated. Several drawbacks are mentioned below.
L11: soil quality is a combination of several soil properties, so probably better to use soil properties here.
L16: “orchards and rainfed agriculture” these two land uses required completely different management, so considered as one unit seems not a proper decision.
L12-16: the abstract should be explanatory and stand alone. Here the most parts of the methods did not mention. So the authors need to include the methods in detail.
L17:” rainfed agriculture” also in orchard field?
L18:” significant changes” include the significant level.
L30-31: please check the journal format for citation.
The introduction section needs to be extensively revised and update to address the hypothesis, methods and compare with the previous studies. Moreover, this section needs to be updated with the recent articles available for similar studies.
https://doi.org/10.1016/j.geoderma.2019.114139
https://doi.org/10.1016/j.ejsobi.2019.103119
Figure 1. include the alphabetic letter for different parts and explain in the caption.
L85-91: it is better to present soil classes with WRB or USDA soil taxonomy.
L107: “LULC” when the first time any abbreviations were used includes the complete name for it.
L110-111: why did not the abb. Include in the Abstract?
L15 and L112: it is confusing, how many soil samples were collected?
Table 2 should be present first then table 1. And merge tables 2 and 3.
Reviewer 3 Report
Dear authors of the manuscript
The manuscript examines soil quality parameters in relation to two different approaches to land use. The parameters of soil fertility have been studied for over a century all over the world. However, obviously, there are still some unclear aspects for the given conditions on a local scale.
Generally, the writing is good, although some improvements can be made in terms of optimizing the repeated information, mis-typing, and grammar.
Please find more detailed comments in the attachment

Reviewer 4 Report
Review report: Sustainability-1494164
-There are several formatting issues with references in the text and also in the reference list. Author should be reformatted it based on the guideline of the journal.
-Please highlight the novelty of this manuscript in the last paragraph of introduction section. What is the novelty of the research in the international context?
-In the method: We all known that soil hydrological properties are regulated by vegetation and soil types. Thus, I suggest you provide a figure of land-cover types.
-When the sampling has been done? Please add it to the text.
-What was the time (duration) between sampling and determination of biological characteristics in the laboratory? Please show it in the related section.
-How many replicates for each assay? The replications should be explained in detail.
-In my opinion the authors should be changed ppm to mg/L or mg/kg in the all text.
Round 2
Reviewer 2 Report
The authors addressed the previous comments, and the revised version improved accordingly. However, some minor comments are listed below:
1- Figure 1 did not cite through the text. Line 110 can be an option.
2- there are several citation format errors " (Error! Reference source not found.)"
3- remove the extra "b" in Fig. 1.
4- for the non-significant cases, please write the correlation coefficients then add "ns".
5-"The maximum value of BD was measured " change to "The maximum value of BD was observed"
Author Response
Please refer attachment
